# γ-Models: Generative Temporal Difference Learning for Infinite-Horizon Prediction

**Michael Janner**[1]     **Igor Mordatch**[2]     **Sergey Levine**[1][2]
[1]UC Berkeley     [2]Google Brain
{janner, svlevine}@eecs.berkeley.edu     imordatch@google.com

## Abstract

We introduce the $\gamma$-model, a predictive model of environment dynamics with an infinite probabilistic horizon. Replacing standard single-step models with $\gamma$-models leads to generalizations of the procedures central to model-based control, including the model rollout and model-based value estimation. The $\gamma$-model, trained with a generative reinterpretation of temporal difference learning, is a natural continuous analogue of the successor representation and a hybrid between model-free and model-based mechanisms. Like a value function, it contains information about the long-term future; like a standard predictive model, it is independent of task reward. We instantiate the $\gamma$-model as both a generative adversarial network and normalizing flow, discuss how its training reflects an inescapable tradeoff between training-time and testing-time compounding errors, and empirically investigate its utility for prediction and control.

## 1   Introduction

The common ingredient in all of model-based reinforcement learning is the dynamics model: a function used for predicting future states. The choice of the model's prediction horizon constitutes a delicate trade-off. Shorter horizons make the prediction problem easier, as the near-term future increasingly begins to look like the present, but may not provide sufficient information for decision-making. Longer horizons carry more information, but present a more difficult prediction problem, as errors accumulate rapidly when a model is applied to its own previous outputs (Janner et al., 2019).

Can we avoid choosing a prediction horizon altogether? Value functions already do so by modeling the cumulative return over a discounted long-term future instead of an immediate reward, circumventing the need to commit to any single finite horizon. However, value prediction folds two problems into one by entangling environment dynamics with reward structure, making value functions less readily adaptable to new tasks in known settings than their model-based counterparts.

In this work, we propose a model that predicts over an infinite horizon with a geometrically-distributed timestep weighting (Figure 1). This $\gamma$-model, named for the dependence of its probabilistic horizon on a discount factor $\gamma$, is trained with a generative analogue of temporal difference learning suitable for continuous spaces. The $\gamma$-model bridges the gap between canonical model-based and model-free mechanisms. Like a value function, it is policy-conditioned and contains information about the distant future; like a conventional dynamics model, it is independent of reward and may be reused for new tasks within the same environment. The $\gamma$-model may be instantiated as both a generative adversarial network (Goodfellow et al., 2014) and a normalizing flow (Rezende & Mohamed, 2015).

The shift from standard single-step models to infinite-horizon $\gamma$-models carries several advantages:

**Constant-time prediction**     Single-step models must perform an $\mathcal{O}(n)$ operation to predict $n$ steps into the future; $\gamma$-models amortize the work of predicting over extended horizons during training such that long-horizon prediction occurs with a single feedforward pass of the model.

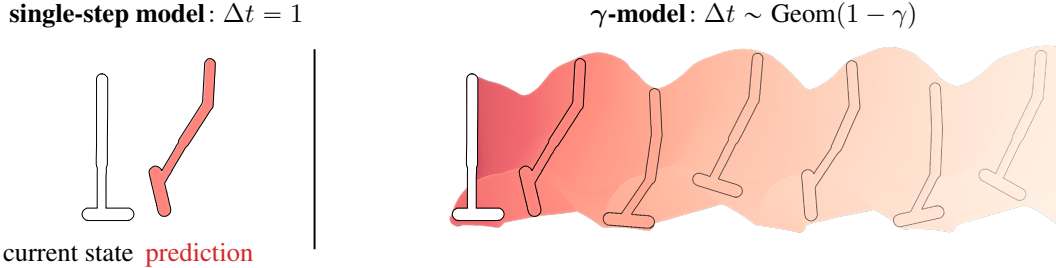

**single-step model**: $\Delta t = 1$        **$\gamma$-model**: $\Delta t \sim \text{Geom}(1 - \gamma)$

current state prediction

Figure 1: Conventional predictive models trained via maximum likelihood have a horizon of one. By interpreting temporal difference learning as a training algorithm for generative models, it is possible to predict with a probabilistic horizon governed by a geometric distribution. In the spirit of infinite-horizon control in model-free reinforcement learning, we refer to this formulation as infinite-horizon prediction.

**Generalized rollouts and value estimation**        Probabilistic prediction horizons lead to generalizations of the core procedures of model-based reinforcement learning. For example, generalized rollouts allow for fine-tuned interpolation between training-time and testing-time compounding error. Similarly, terminal value functions appended to truncated $\gamma$-model rollouts allow for a gradual transition between model-based and model-free value estimation.

**Omission of unnecessary information**        The predictions of a $\gamma$-model do not come paired with an associated timestep. While on the surface a limitation, we show why knowing precisely *when* a state will be encountered is not necessary for decision-making. Infinite-horizon $\gamma$-model prediction selectively discards the unnecessary information from a standard model-based rollout.

## 2   Related Work

The complementary strengths and weaknesses of model-based and model-free reinforcement learning have led to a number of works that attempt to combine these approaches. Common strategies include initializing a model-free algorithm with the solution found by a model-based planner (Levine & Koltun, 2013; Farshidian et al., 2014; Nagabandi et al., 2018), feeding model-generated data into an otherwise model-free optimizer (Sutton, 1990; Silver et al., 2008; Lampe & Riedmiller, 2014; Kalweit & Boedecker, 2017; Luo et al., 2019), using model predictions to improve the quality of target values for temporal difference learning (Buckman et al., 2018; Feinberg et al., 2018), leveraging model gradients for backpropagation (Nguyen & Widrow, 1990; Jordan & Rumelhart, 1992; Heess et al., 2015), and incorporating model-based planning without explicitly predicting future observations (Tamar et al., 2016; Silver et al., 2017; Oh et al., 2017; Kahn et al., 2018; Amos et al., 2018; Schrittwieser et al., 2019). In contrast to combining independent model-free and model-based components, we describe a framework for training a new class of predictive model with a generative, model-based reinterpretation of model-free tools.

Temporal difference models (TDMs) Pong et al. (2018) provide an alternative method of training models with what are normally considered to be model-free algorithms. TDMs interpret models as a special case of goal-conditioned value functions (Kaelbling, 1993; Foster & Dayan, 2002; Schaul et al., 2015; Andrychowicz et al., 2017), though the TDM is constrained to predict at a fixed horizon and is limited to tasks for which the reward depends only on the last state. In contrast, the $\gamma$-model predicts over a discounted infinite-horizon future and accommodates arbitrary rewards.

The most closely related prior work to $\gamma$-models is the successor representation (Dayan, 1993), a formulation of long-horizon prediction that has been influential in both cognitive science (Momennejad et al., 2017; Gershman, 2018) and machine learning (Kulkarni et al., 2016; Ma et al., 2018). In its original form, the successor representation is tractable only in tabular domains. Prior continuous variants have focused on policy evaluation based on expected state featurizations (Barreto et al., 2017, 2018; Hansen et al., 2020), forgoing an interpretation as a probabilistic model suitable for state prediction. Converting the tabular successor representation into a continuous generative model is non-trivial because the successor representation implicitly assumes the ability to normalize over a finite state space for interpretation as a predictive model.

Because of the discounted state occupancy's central role in reinforcement learning, its approximation by Bellman equations has been the focus of multiple lines of work. Generalizations include $\beta$-models

(Sutton, 1995), allowing for arbitrary mixture distributions over time, and option models (Sutton et al., 1999), allowing for state-dependent termination conditions. While our focus is on generative models featuring the state-independent geometric timestep weighting of the successor representation, we are hopeful that the tools developed in this paper could also be applicable in the design of continuous analogues of these generalizations.

## 3 Preliminaries

We consider an infinite-horizon Markov decision process (MDP) defined by the tuple $(\mathcal{S}, \mathcal{A}, p, r, \gamma)$, with state space $\mathcal{S}$ and action space $\mathcal{A}$. The transition distribution and reward function are given by $p : \mathcal{S} \times \mathcal{A} \times \mathcal{S} \to \mathbb{R}^+$ and $r : \mathcal{S} \to \mathbb{R}$, respectively. The discount is denoted by $\gamma \in (0, 1)$. A policy $\pi : \mathcal{S} \times \mathcal{A} \to \mathbb{R}^+$ induces a conditional occupancy $\mu(\mathbf{s} \mid \mathbf{s}_t, \mathbf{a}_t)$ over future states:

$$\mu(\mathbf{s} \mid \mathbf{s}_t, \mathbf{a}_t) = (1 - \gamma) \sum_{\Delta t = 1}^{\infty} \gamma^{\Delta t - 1} p(\mathbf{s}_{t+\Delta t} = \mathbf{s} \mid \mathbf{s}_t, \mathbf{a}_t, \pi). \tag{1}$$

We denote parametric approximations of $p$ ($\mu$) as $p_\theta$ ($\mu_\theta$), in which the subscripts denote model parameters. Standard model-based reinforcement learning algorithms employ the single-step model $p_\theta$ for long-horizon decision-making by performing multi-step model-based rollouts.

## 4 Generative Temporal Difference Learning

Our goal is to make long-horizon predictions without the need to repeatedly apply a single-step model. Instead of modeling states at a particular instant in time by approximating the environment transition distribution $p(\mathbf{s}_{t+1} \mid \mathbf{s}_t, \mathbf{a}_t)$, we aim to predict a weighted distribution over all possible future states according to $\mu(\mathbf{s} \mid \mathbf{s}_t, \mathbf{a}_t)$. In principle, this can be posed as a conventional maximum likelihood problem:

$$\max_\theta \ \mathbb{E}_{\mathbf{s}_t, \mathbf{a}_t, \mathbf{s} \sim \mu(\cdot \mid \mathbf{s}_t, \mathbf{a}_t)} \left[ \log \mu_\theta(\mathbf{s} \mid \mathbf{s}_t, \mathbf{a}_t) \right].$$

However, doing so would require collecting samples from the occupancy $\mu$ independently for each policy of interest. Forgoing the ability to re-use data from multiple policies when training dynamics models would sacrifice the sample efficiency that often makes model usage compelling in the first place, so we instead aim to design an off-policy algorithm for training $\mu_\theta$. We accomplish this by reinterpreting temporal difference learning as a method for training generative models.

Instead of collecting only on-policy samples from $\mu(\mathbf{s} \mid \mathbf{s}_t, \mathbf{a}_t)$, we observe that $\mu$ admits a convenient recursive form. Consider a modified MDP in which there is a $1 - \gamma$ probability of terminating at each timestep. The distribution over the state at termination, denoted as the exit state $\mathbf{s}_e$, corresponds to first sampling from a termination timestep $\Delta t \sim \text{Geom}(1 - \gamma)$ and then sampling from the per-timestep distribution $p(\mathbf{s}_{t+\Delta t} \mid \mathbf{s}_t, \mathbf{a}_t, \pi)$. The distribution over $\mathbf{s}_e$ corresponds exactly to that in the definition of the occupancy $\mu$ in Equation 1, but also lends itself to an interpretation as a mixture over only two components: the distribution at the immediate next timestep, in the event of termination, and that over all subsequent timesteps, in the event of non-termination. This mixture yields the following target distribution:

$$p_{\text{targ}}(\mathbf{s}_e \mid \mathbf{s}_t, \mathbf{a}_t) = \underbrace{(1 - \gamma) p(\mathbf{s}_e \mid \mathbf{s}_t, \mathbf{a}_t)}_{\text{single-step distribution}} + \underbrace{\gamma \mathbb{E}_{\mathbf{s}_{t+1} \sim p(\cdot \mid \mathbf{s}_t, \mathbf{a}_t)} \left[ \mu_\theta(\mathbf{s}_e \mid \mathbf{s}_{t+1}) \right]}_{\text{model bootstrap}}. \tag{2}$$

We use the shorthand $\mu_\theta(\mathbf{s}_e \mid \mathbf{s}_{t+1}) = \mathbb{E}_{\mathbf{a}_{t+1} \sim \pi(\cdot \mid \mathbf{s}_{t+1})} \left[ \mu_\theta(\mathbf{s}_e \mid \mathbf{s}_{t+1}, \mathbf{a}_{t+1}) \right]$. The target distribution $p_{\text{targ}}$ is reminiscent of a temporal difference target value: the state-action conditioned occupancy $\mu_\theta(\mathbf{s}_e \mid \mathbf{s}_t, \mathbf{a}_t)$ acts as a $Q$-function, the state-conditioned occupancy $\mu_\theta(\mathbf{s}_e \mid \mathbf{s}_{t+1})$ acts as a value function, and the single-step distribution $p(\mathbf{s}_{t+1} \mid \mathbf{s}_t, \mathbf{a}_t)$ acts as a reward function. However, instead of representing a scalar target value, $p_{\text{targ}}$ is a distribution from which we may sample future states $\mathbf{s}_e$. We can use this target distribution in place of samples from the true discounted occupancy $\mu$:

$$\max_\theta \ \mathbb{E}_{\mathbf{s}_t, \mathbf{a}_t, \mathbf{s}_e \sim (1-\gamma) p(\cdot \mid \mathbf{s}_t, \mathbf{a}_t) + \gamma \mathbb{E}[\mu_\theta(\cdot \mid \mathbf{s}_{t+1})]} \left[ \log \mu_\theta(\mathbf{s}_e \mid \mathbf{s}_t, \mathbf{a}_t) \right].$$

This formulation differs from a standard maximum likelihood learning problem in that the target distribution depends on the current model. By bootstrapping the target distribution in this manner, we

are able to use only empirical $(\mathbf{s}_t, \mathbf{a}_t, \mathbf{s}_{t+1})$ transitions from one policy in order to train an infinite-horizon predictive model $\mu_\theta$ for any other policy. Because the horizon is governed by the discount $\gamma$, we refer to such a model as a $\gamma$-model.

This bootstrapped model training may be incorporated into a number of different generative modeling frameworks. We discuss two cases here. (1) When the model $\mu_\theta$ permits only sampling, we may train $\mu_\theta$ by minimizing an $f$-divergence from samples:

$$\mathcal{L}_1(\mathbf{s}_t, \mathbf{a}_t, \mathbf{s}_{t+1}) = D_f(\mu_\theta(\cdot \mid \mathbf{s}_t, \mathbf{a}_t) \,\|\, (1-\gamma)p(\cdot \mid \mathbf{s}_t, \mathbf{a}_t) + \gamma\mu_\theta(\cdot \mid \mathbf{s}_{t+1})). \tag{3}$$

This objective leads naturally to an adversarially-trained $\gamma$-model. (2) When the model $\mu_\theta$ permits density evaluation, we may minimize an error defined on log-densities directly:

$$\mathcal{L}_2(\mathbf{s}_t, \mathbf{a}_t, \mathbf{s}_{t+1}) = \mathbb{E}_{\mathbf{s}_e}\Big[\big\|\log\mu_\theta(\mathbf{s}_e \mid \mathbf{s}_t, \mathbf{a}_t) - \log\big((1-\gamma)p(\mathbf{s}_e \mid \mathbf{s}_t, \mathbf{a}_t) + \gamma\mu_\theta(\mathbf{s}_e \mid \mathbf{s}_{t+1})\big)\big\|_2^2\Big]. \tag{4}$$

This objective is suitable for $\gamma$-models instantiated as normalizing flows. Due to the approximation of a target log-density $\log\big((1-\gamma)p(\cdot \mid \mathbf{s}_t, \mathbf{a}_t) + \gamma\mathbb{E}_{\mathbf{s}_{t+1}}[\mu_\theta(\cdot \mid \mathbf{s}_{t+1})]\big)$ using a single next state $\mathbf{s}_{t+1}$, $\mathcal{L}_2$ is unbiased for deterministic dynamics and a bound in the case of stochastic dynamics. We provide complete algorithmic descriptions of both variants and highlight practical training considerations in Section 6.

## 5 Analysis and Applications of $\gamma$-Models

Using the $\gamma$-model for prediction and control requires us to generalize procedures common in model-based reinforcement learning. In this section, we derive the $\gamma$-model rollout and show how it can be incorporated into a reinforcement learning procedure that hybridizes model-based and model-free value estimation. First, however, we show that the $\gamma$-model is a continuous, generative counterpart to another type of long-horizon model: the successor representation.

### 5.1 $\gamma$-Models as a Continuous Successor Representation

The successor representation $M$ is a prediction of expected visitation counts (Dayan, 1993). It has a recurrence relation making it amenable to tabular temporal difference algorithms:

$$M(\mathbf{s}_e \mid \mathbf{s}_t, \mathbf{a}_t) = \mathbb{E}_{\mathbf{s}_{t+1}\sim p(\cdot \mid \mathbf{s}_t, \mathbf{a}_t)}\left[\mathbb{1}\left[\mathbf{s}_e = \mathbf{s}_{t+1}\right] + \gamma M(\mathbf{s}_e \mid \mathbf{s}_{t+1})\right]. \tag{5}$$

Adapting the successor representation to continuous state spaces in a way that retains an interpretation as a probabilistic model has proven challenging. However, variants that forego the ability to sample in favor of estimating expected state features have been developed (Barreto et al., 2017).

The form of the successor recurrence relation bears a striking resemblance to that of the target distribution in Equation 2, suggesting a connection between the generative, continuous $\gamma$-model and the discriminative, tabular successor representation. We now make this connection precise.

**Proposition 1.** *The global minimum of both $\mathcal{L}_1$ and $\mathcal{L}_2$ is achieved if and only if the resulting $\gamma$-model produces samples according to the normalized successor representation:*

$$\mu_\theta(\mathbf{s}_e \mid \mathbf{s}_t, \mathbf{a}_t) = (1-\gamma)M(\mathbf{s}_e \mid \mathbf{s}_t, \mathbf{a}_t).$$

*Proof.* In the case of either objective, the global minimum is achieved only when

$$\mu_\theta(\mathbf{s}_e \mid \mathbf{s}_t, \mathbf{a}_t) = (1-\gamma)p(\mathbf{s}_e \mid \mathbf{s}_t, \mathbf{a}_t) + \gamma\mathbb{E}_{\mathbf{s}_{t+1}\sim p(\cdot \mid \mathbf{s}_t, \mathbf{a}_t)}\left[\mu_\theta(\mathbf{s}_e \mid \mathbf{s}_{t+1})\right]$$

for all $\mathbf{s}_t, \mathbf{a}_t$. We recognize this optimality condition exactly as the recurrence defining the successor representation $M$ (Equation 5), scaled by $(1-\gamma)$ such that $\mu_\theta$ integrates to 1 over $\mathbf{s}_e$. $\qquad\square$

### 5.2 $\gamma$-Model Rollouts

Standard single-step models, which correspond to $\gamma$-models with $\gamma = 0$, can predict multiple steps into the future by making iterated autoregressive predictions, conditioning each step on their own output from the previous step. These sequential rollouts form the foundation of most model-based

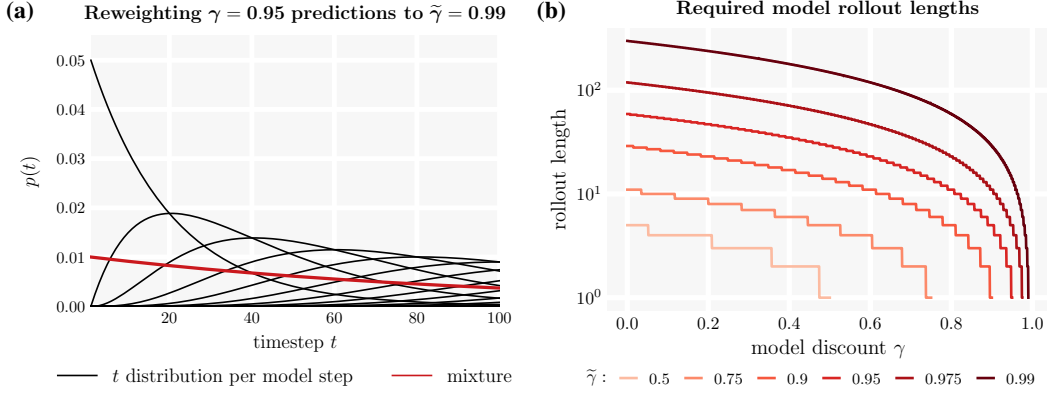

Figure 2: **(a)** The first step from a $\gamma$-model samples states at timesteps distributed according to a geometric distribution with parameter $1 - \gamma$; all subsequent steps have a negative binomial timestep distribution stemming from the sum of independent geometric random variables. When these steps are reweighted according to Theorem 1, the resulting distribution follows a geometric distribution with smaller parameter (corresponding to a larger discount value $\tilde{\gamma}$). **(b)** The number of steps needed to recover 95% of the probability mass from distributions induced by various target discounts $\tilde{\gamma}$ for all valid model discounts $\gamma$. When using a standard single-step model, corresponding to the case of $\gamma = 0$, a 299-step model rollout is required to reweight to a discount of $\tilde{\gamma} = 0.99$.

reinforcement learning algorithms. We now generalize these rollouts to $\gamma$-models for $\gamma > 0$, allowing us to decouple the discount used during model training from the desired horizon in control. When working with multiple discount factors, we explicitly condition an occupancy on its discount as $\mu(\mathbf{s}_e \mid \mathbf{s}_t; \gamma)$. In the results below, we omit the model parameterization $\theta$ whenever a statement applies to both a discounted occupancy $\mu$ and a parametric $\gamma$-model $\mu_\theta$.

**Theorem 1.** *Let $\mu_n(\mathbf{s}_e \mid \mathbf{s}_t; \gamma)$ denote the distribution over states at the $n^{th}$ sequential step of a $\gamma$-model rollout beginning from state $\mathbf{s}_t$. For any desired discount $\tilde{\gamma} \in [\gamma, 1)$, we may reweight the samples from these model rollouts according to the weights*

$$\alpha_n = \frac{(1 - \tilde{\gamma})(\tilde{\gamma} - \gamma)^{n-1}}{(1 - \gamma)^n}$$

*to obtain the state distribution drawn from $\mu_1(\mathbf{s}_e \mid \mathbf{s}_t; \tilde{\gamma}) = \mu(\mathbf{s}_e \mid \mathbf{s}_t; \tilde{\gamma})$. That is, we may reweight the steps of a $\gamma$-model rollout so as to match the distribution of a $\tilde{\gamma}$-model with larger discount:*

$$\mu(\mathbf{s}_e \mid \mathbf{s}_t; \tilde{\gamma}) = \sum_{n=1}^{\infty} \alpha_n \mu_n(\mathbf{s}_e \mid \mathbf{s}_t; \gamma).$$

*Proof.* See Appendix A. □

This reweighting scheme has two special cases of interest. A standard single-step model, with $\gamma = 0$, yields $\alpha_n = (1 - \tilde{\gamma})\tilde{\gamma}^{n-1}$. These weights are familiar from the definition of the discounted state occupancy in terms of a per-timestep mixture (Equation 1). Setting $\gamma = \tilde{\gamma}$ yields $\alpha_n = 0^{n-1}$, or a weight of 1 on the first step and 0 on all subsequent steps.[1] This result is also expected: when the model discount matches the target discount, only a single forward pass of the model is required.

Figure 2 visually depicts the reweighting scheme and the number of steps required for truncated model rollouts to approximate the distribution induced by a larger discount. There is a natural tradeoff with $\gamma$-models: the higher $\gamma$ is, the fewer model steps are needed to make long-horizon predictions, reducing model-based compounding prediction errors (Asadi et al., 2019; Janner et al., 2019). However, increasing $\gamma$ transforms what would normally be a standard maximum likelihood problem (in the case of single-step models) into one resembling approximate dynamic programming (with a model bootstrap), leading to model-free bootstrap error accumulation (Kumar et al., 2019). The primary distinction is whether this accumulation occurs during training, when the work of sampling from the occupancy $\mu$ is being amortized, or during "testing", when the model is being used for rollouts. While this horizon-based error compounding cannot be eliminated entirely, $\gamma$-models allow for a continuous interpolation between the two extremes.

## 5.3 γ-Model-Based Value Expansion

We now turn our attention from prediction with $\gamma$-models to value estimation for control. In tabular domains, the state-action value function can be decomposed as the inner product between the successor representation $M$ and the vector of per-state rewards (Gershman, 2018). Taking care to account for the normalization from the equivalence in Proposition 1, we can similarly estimate the $Q$ function as the expectation of reward under states sampled from the $\gamma$-model:

$$
\begin{aligned}
Q(\mathbf{s}_t, \mathbf{a}_t; \gamma) &= \sum_{\Delta t=1}^{\infty} \gamma^{\Delta t-1} \int_{\mathcal{S}} r(\mathbf{s}_e) p(\mathbf{s}_{t+\Delta t} = \mathbf{s}_e \mid \mathbf{s}_t, \mathbf{a}_t, \pi) \mathrm{d}\mathbf{s}_e \\
&= \int_{\mathcal{S}} r(\mathbf{s}_e) \sum_{\Delta t=1}^{\infty} \gamma^{\Delta t-1} p(\mathbf{s}_{t+\Delta t} = \mathbf{s}_e \mid \mathbf{s}_t, \mathbf{a}_t, \pi) \mathrm{d}\mathbf{s}_e \\
&= \frac{1}{1-\gamma} \mathbb{E}_{\mathbf{s}_e \sim \mu(\cdot|\mathbf{s}_t, \mathbf{a}_t; \gamma)} \left[ r(\mathbf{s}_e) \right]
\end{aligned}
\tag{6}
$$

This relation suggests a model-based reinforcement learning algorithm in which $Q$-values are estimated by a $\gamma$-model without the need for sequential model-based rollouts. However, in some cases it may be practically difficult to train a generative $\gamma$-model with discount as large as that of a discriminative $Q$-function. While one option is to chain together $\gamma$-model steps as in Section 5.2, an alternative solution often effective with single-step models is to combine short-term value estimates from a truncated model rollout with a terminal model-free value prediction:

$$
V_{\mathrm{MVE}}(\mathbf{s}_t; \tilde{\gamma}) = \sum_{n=1}^{H} \tilde{\gamma}^{n-1} r(\mathbf{s}_{t+n}) + \tilde{\gamma}^{H} V(\mathbf{s}_{t+H}; \tilde{\gamma}).
$$

This hybrid estimator is referred to as a model-based value expansion (MVE; Feinberg et al. 2018). There is a hard transition between the model-based and model-free value estimation in MVE, occuring at the model horizon $H$. We may replace the single-step model with a $\gamma$-model for a similar estimator in which there is a probabilistic prediction horizon, and as a result a gradual transition:

$$
V_{\gamma\text{-MVE}}(\mathbf{s}_t; \tilde{\gamma}) = \frac{1}{1-\tilde{\gamma}} \sum_{n=1}^{H} \alpha_n \mathbb{E}_{\mathbf{s}_e \sim \mu_n(\cdot|\mathbf{s}_t; \gamma)} \left[ r(\mathbf{s}_e) \right] + \left( \frac{\tilde{\gamma}-\gamma}{1-\gamma} \right)^{H} \mathbb{E}_{\mathbf{s}_e \sim \mu_H(\cdot|\mathbf{s}_t; \gamma)} \left[ V(\mathbf{s}_e; \tilde{\gamma}) \right].
$$

The $\gamma$-MVE estimator allows us to perform $\gamma$-model-based rollouts with horizon $H$, reweight the samples from this rollout by solving for weights $\alpha_n$ given a desired discount $\tilde{\gamma} > \gamma$, and correct for the truncation error stemming from the finite rollout length using a terminal value function with discount $\tilde{\gamma}$. As expected, MVE is a special case of $\gamma$-MVE, as can be verified by considering the weights corresponding to $\gamma = 0$ described in Section 5.2. This estimator, along with the simpler value estimation in Equation 6, highlights the fact that it is not necessary to have timesteps associated with states in order to use predictions for decision-making. We provide a more thorough treatment of $\gamma$-MVE, complete with pseudocode for a corresponding actor-critic algorithm, in Appendix B.

## 6 Practical Training of γ-Models

Because $\gamma$-model training differs from standard dynamics modeling primarily in the bootstrapped target distribution and not in the model parameterization, $\gamma$-models are in principle compatible with any generative modeling framework. We focus on two representative scenarios, differing in whether the generative model class used to instantiate the $\gamma$-model allows for tractable density evaluation.

**Training without density evaluation**    When the $\gamma$-model parameterization does not allow for tractable density evaluation, we minimize a bootstrapped $f$-divergence according to $\mathcal{L}_1$ (Equation 3) using only samples from the model. The generative adversarial framework provides a convenient way to train a parametric generator by minimizing an $f$-divergence of choice given only samples from a target distribution $p_{\mathrm{targ}}$ and the ability to sample from the generator (Goodfellow et al., 2014; Nowozin et al., 2016). In the case of bootstrapped maximum likelihood problems, our target distribution is induced by the model itself (alongside a single-step transition distribution), meaning that we only need sample access to our $\gamma$-model in order to train $\mu_\theta$ as a generative adversarial network (GAN).

Introducing an auxiliary discriminator $D_\phi$ and selecting the Jensen-Shannon divergence as our $f$-divergence, we can reformulate minimization of the original objective $\mathcal{L}_1$ as a saddle-point

---

**Algorithm 1** $\gamma$-model training without density evaluation

---
1: **Input** $\mathcal{D}$ : dataset of transitions, $\pi$ : policy, $\lambda$ : step size, $\tau$ : delay parameter
2: Initialize parameter vectors $\theta, \bar{\theta}, \phi$
3: **while** not converged **do**
4:     Sample transitions $(\mathbf{s}_t, \mathbf{a}_t, \mathbf{s}_{t+1})$ from $\mathcal{D}$ and actions $\mathbf{a}_{t+1} \sim \pi(\cdot \mid \mathbf{s}_{t+1})$
5:     Sample from bootstrapped target $\mathbf{s}_e^+ \sim (1-\gamma)\delta_{\mathbf{s}_{t+1}} + \gamma\mu_{\bar{\theta}}(\cdot \mid \mathbf{s}_{t+1}, \mathbf{a}_{t+1})$
6:     Sample from current model $\mathbf{s}_e^- \sim \mu_\theta(\cdot \mid \mathbf{s}_t, \mathbf{a}_t)$
7:     Evaluate objective $\mathcal{L} = \log D_\phi(\mathbf{s}_e^+ \mid \mathbf{s}_t, \mathbf{a}_t) + \log(1 - D_\phi(\mathbf{s}_e^- \mid \mathbf{s}_t, \mathbf{a}_t))$
8:     Update model parameters $\theta \leftarrow \theta - \lambda\nabla_\theta\mathcal{L}$; $\phi \leftarrow \phi + \lambda\nabla_\phi\mathcal{L}$
9:     Update target parameters $\bar{\theta} \leftarrow \tau\theta + (1-\tau)\bar{\theta}$
10: **end while**

---

**Algorithm 2** $\gamma$-model training with density evaluation

---
1: **Input** $\mathcal{D}$ : dataset of transitions, $\pi$ : policy, $\lambda$ : step size, $\tau$ : delay parameter, $\sigma^2$ : variance
2: Initialize parameter vectors $\theta, \bar{\theta}$; let $f$ denote the Gaussian pdf
3: **while** not converged **do**
4:     Sample transitions $(\mathbf{s}_t, \mathbf{a}_t, \mathbf{s}_{t+1})$ from $\mathcal{D}$ and actions $\mathbf{a}_{t+1} \sim \pi(\cdot \mid \mathbf{s}_{t+1})$
5:     Sample from bootstrapped target $\mathbf{s}_e \sim (1-\gamma)\mathcal{N}(\mathbf{s}_{t+1}, \sigma^2) + \gamma\mu_{\bar{\theta}}(\cdot \mid \mathbf{s}_{t+1}, \mathbf{a}_{t+1})$
6:     Construct target values $T = \log\left((1-\gamma)f(\mathbf{s}_e \mid \mathbf{s}_{t+1}, \sigma^2) + \gamma\mu_{\bar{\theta}}(\mathbf{s}_e \mid \mathbf{s}_{t+1}, \mathbf{a}_{t+1})\right)$
7:     Evaluate objective $\mathcal{L} = \|\log \mu_\theta(\mathbf{s}_e \mid \mathbf{s}_t, \mathbf{a}_t) - T\|_2^2$
8:     Update model parameters $\theta \leftarrow \theta - \lambda\nabla_\theta\mathcal{L}$
9:     Update target parameters $\bar{\theta} \leftarrow \tau\theta + (1-\tau)\bar{\theta}$
10: **end while**

---

optimization over the following objective:

$$\hat{\mathcal{L}}_1(\mathbf{s}_t, \mathbf{a}_t) = \mathbb{E}_{\mathbf{s}_e^+ \sim p_{\text{targ}}(\cdot|\mathbf{s}_t,\mathbf{a}_t)}\left[\log D_\phi(\mathbf{s}_e^+ \mid \mathbf{s}_t, \mathbf{a}_t)\right] + \mathbb{E}_{\mathbf{s}_e^- \sim \mu_\theta(\cdot|\mathbf{s}_t,\mathbf{a}_t)}\left[\log(1 - D_\phi(\mathbf{s}_e^- \mid \mathbf{s}_t, \mathbf{a}_t))\right],$$

which is minimized over $\mu_\theta$ and maximized over $D_\phi$. As in $\mathcal{L}_1$, $p_{\text{targ}}$ refers to the bootstrapped target distribution in Equation 2. In this formulation, $\mu_\theta$ produces samples by virtue of a deterministic mapping of a random input vector $\mathbf{z} \sim \mathcal{N}(0, I)$ and conditioning information $(\mathbf{s}_t, \mathbf{a}_t)$. Other choices of $f$-divergence may be instantiated by different choices of activation function (Nowozin et al., 2016).

**Training with density evaluation** When the $\gamma$-model permits density evaluation, we may bypass saddle point approximations to an $f$-divergence and directly regress to target density values, as in objective $\mathcal{L}_2$ (Equation 4). This is a natural choice when the $\gamma$-model is instantiated as a conditional normalizing flow (Rezende & Mohamed, 2015). Evaluating target values of the form

$$T(\mathbf{s}_t, \mathbf{a}_t, \mathbf{s}_{t+1}, \mathbf{s}_e) = \log\left((1-\gamma)p(\mathbf{s}_e \mid \mathbf{s}_t, \mathbf{a}_t) + \gamma\mu_\theta(\mathbf{s}_e \mid \mathbf{s}_{t+1})\right)$$

requires density evaluation of not only our $\gamma$-model, but also the single-step transition distribution. There are two options for estimating the single-step densities: (1) a single-step model $p_\theta$ may be trained alongside the $\gamma$-model $\mu_\theta$ for the purposes of constructing targets $T(\mathbf{s}_t, \mathbf{a}_t, \mathbf{s}_{t+1}, \mathbf{s}_e)$, or (2) a simple approximate model may be constructed on the fly from $(\mathbf{s}_t, \mathbf{a}_t, \mathbf{s}_{t+1})$ transitions. We found $p_\theta = \mathcal{N}(\mathbf{s}_{t+1}, \sigma^2)$, with $\sigma^2$ a constant hyperparameter, to be sufficient.

**Stability considerations** To alleviate the instability caused by bootstrapping, we appeal to the standard solution employed in model-free reinforcement learning: decoupling the regression targets from the current model by way of a "delayed" target network (Mnih et al., 2015). In particular, we use a delayed $\gamma$-model $\mu_{\bar{\theta}}$ in the bootstrapped target distribution $p_{\text{targ}}$, with the parameters $\bar{\theta}$ given by an exponentially-moving average of previous parameters $\theta$.

We summarize the above scenarios in Algorithms 1 and 2. We isolate model training from data collection and focus on a setting in which a static dataset is provided, but this algorithm may also be used in a data-collection loop for policy improvement. Further implementation details, including all hyperparameter settings and network architectures, are included in Appendix C.

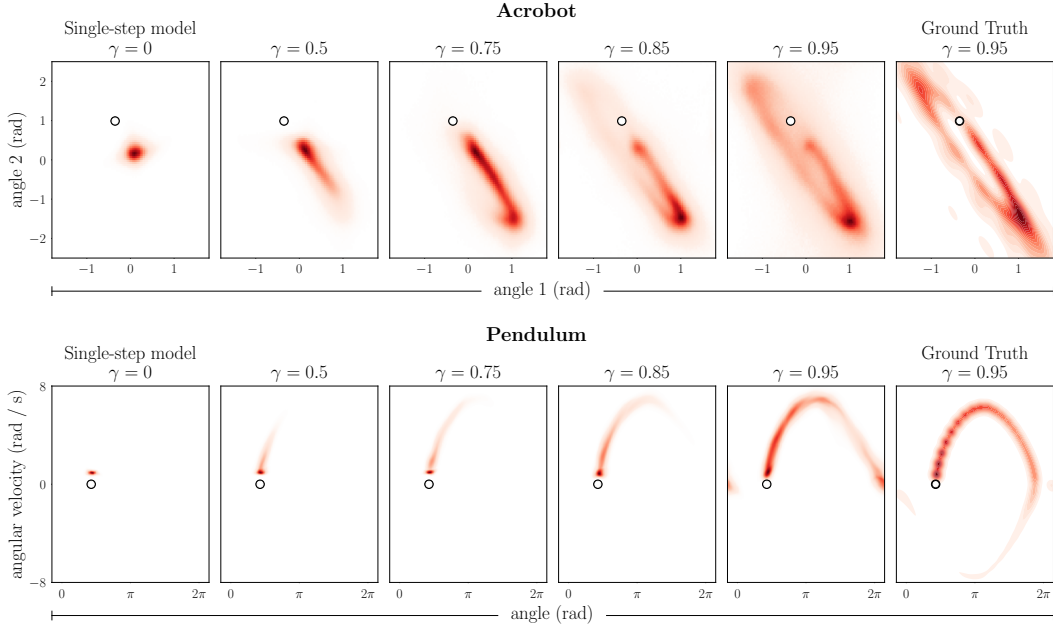

Figure 3: Visualization of the predicted distribution from a **single** feedforward pass of normalizing flow $\gamma$-models trained with varying discounts $\gamma$. The conditioning state $\mathbf{s}_t$ is denoted by $\circ$. The leftmost plots, with $\gamma = 0$, correspond to a single-step model. For comparison, the rightmost plots show a Monte Carlo estimation of the discounted occupancy from 100 environment trajectories.

## 7 Experiments

Our experimental evaluation is designed to study the viability of $\gamma$-models as a replacement of conventional single-step models for long-horizon state prediction and model-based control.

### 7.1 Prediction

We investigate $\gamma$-model predictions as a function of discount in continuous-action versions of two benchmark environments suitable for visualization: acrobot (Sutton, 1996) and pendulum. The training data come from a mixture distribution over all intermediate policies of 200 epochs of optimization with soft-actor critic (SAC; Haarnoja et al. 2018). The final converged policy is used for $\gamma$-model training. We refer to Appendix C for implementation and experiment details.

Figure 3 shows the predictions of a $\gamma$-model trained as a normalizing flow according to Algorithm 2 for five different discounts, ranging from $\gamma = 0$ (a single-step model) to $\gamma = 0.95$. The rightmost column shows the ground truth discounted occupancy corresponding to $\gamma = 0.95$, estimated with Monte Carlo rollouts of the policy. Increasing the discount $\gamma$ during training has the expected effect of qualitatively increasing the predictive lookahead of a single feedforward pass of the $\gamma$-model. We found flow-based $\gamma$-models to be more reliable than GAN parameterizations, especially at higher discounts. Corresponding GAN $\gamma$-model visualizations can be found in Appendix E for comparison.

Equation 6 expresses values as an expectation over a single feedforward pass of a $\gamma$-model. We visualize this relation in Figure 4, which depicts $\gamma$-model predictions on the pendulum environment for a discount of $\gamma = 0.99$ and the resulting value map estimated by taking expectations over these predicted state distributions. In comparison, value estimation for the same discount using a single-step model would require 299-step rollouts in order to recover $95\%$ of the probability mass (see Figure 2).

### 7.2 Control

To study the utility of the $\gamma$-model for model-based reinforcement learning, we use the $\gamma$-MVE estimator from Section 5.3 as a drop-in replacement for value estimation in SAC. We compare this approach to the state-of-the-art in model-based and model-free methods, with representative algorithms consisting of SAC, PPO (Schulman et al., 2017), MBPO (Janner et al., 2019), and MVE (Feinberg et al., 2018). In $\gamma$-MVE, we use a model discount of $\gamma = 0.8$, a value discount of

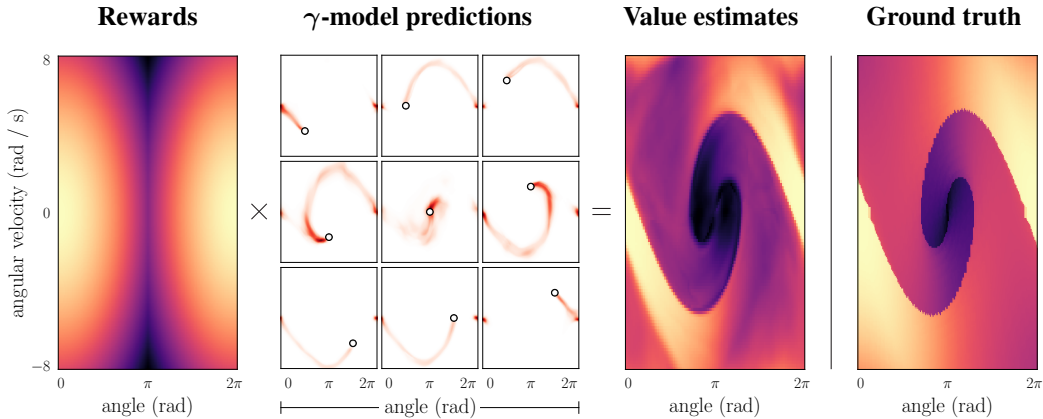

Figure 4: Values are expectations of reward over a single feedforward pass of a $\gamma$-model (Equation 6). We visualize $\gamma$-model predictions ($\gamma = 0.99$) from nine starting states, denoted by $\circ$, in the pendulum benchmark environment. Taking the expectation of reward over each of these predicted distributions yields a value estimate for the corresponding conditioning state. The rightmost plot depicts the value map produced by value iteration on a discretization of the same environment for reference.

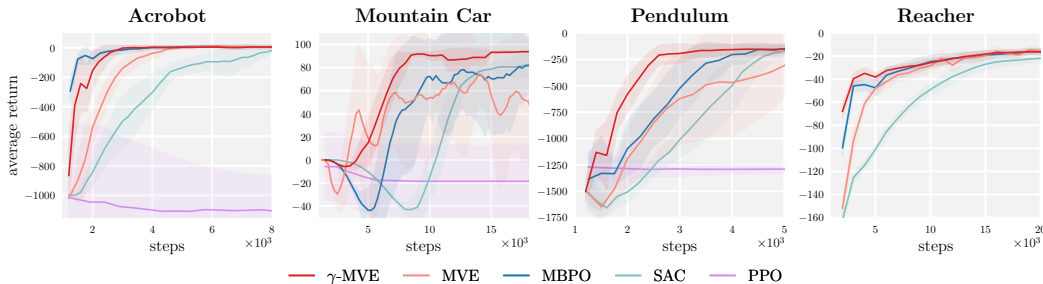

Figure 5: Comparative performance of $\gamma$-MVE and four prior reinforcement learning algorithms on continuous control benchmark tasks. $\gamma$-MVE retains the asymptotic performance of SAC with sample-efficiency matching that of MBPO. Shaded regions depict standard deviation among 5 seeds.

$\tilde{\gamma} = 0.99$ and a single model step ($n = 1$). We use a model rollout length of 5 in MVE such that it has an effective horizon identical to that of $\gamma$-MVE. Other hyperparameter settings can once again be found in Appendix C; details regarding the evaluation environments can be found in Appendix D. Figure 5 shows learning curves for all methods. We find that $\gamma$-MVE converges faster than prior algorithms, twice as quickly as SAC, while retaining their asymptotic performance.

# 8   Discussion, Limitations, and Future Work

We have introduced a new class of predictive model, a $\gamma$-model, that is a hybrid between standard model-free and model-based mechanisms. It is policy-conditioned and infinite-horizon, like a value function, but independent of reward, like a standard single-step model. This new formulation of infinite-horizon prediction allows us to generalize the procedures integral to model-based control, yielding new variants of model rollouts and model-based value estimation.

Our experimental evaluation shows that, on tasks with low to moderate dimensionality, our method learns accurate long-horizon predictive distributions without sequential rollouts and can be incorporated into standard model-based reinforcement learning methods to produce results that are competitive with state-of-the-art algorithms. Scaling up our framework to more complex tasks, including high-dimensional continuous control problems and tasks with image observations, presents a number of additional challenges. We are optimistic that continued improvements in training techniques for generative models and increased stability in off-policy reinforcement learning will also carry benefits for $\gamma$-model training.

## Broader Impact

$\gamma$-models provide a way to study infinite-horizon prediction in the same way that we can study the problem of control in infinite-horizon MDPs, possibly providing a path for more accurate long-term modeling without test-time compounding error. However, transforming a maximum likelihood problem into an approximate dynamic programming one is not without its drawbacks: maximum likelihood learning techniques are substantially better understood at this point in time and can be accompanied by rigorous guarantees even when incorporating function approximation. It will take much future work not just on $\gamma$-models, but on dynamic programming methods more broadly, for them to work reliably enough for deployment in real-world, safety-critical systems. While much work remains, we are optimistic for the long-term viability of temporal difference learning as an algorithm for training long-horizon dynamics models given its empirical success in long-horizon model-free control.

### Acknowledgements

We thank Karthik Narasimhan, Pulkit Agrawal, Anirudh Goyal, Pedro Tsividis, Anusha Nagabandi, Aviral Kumar, and Michael Chang for formative discussions about model-based reinforcement learning and generative modeling. This work was partially supported by computational resource donations from Amazon. M.J. is supported by fellowships from the National Science Foundation and the Open Philanthropy Project.

## Footnotes

[1] We define $0^0$ as $\lim_{x \to 0} x^x = 1$.

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
