[Supplementary Material]

# Appendix A  Derivation of $\gamma$-Model-Based Rollout Weights

**Theorem 1.** *Let $\mu_n(\mathbf{s}_e \mid \mathbf{s}_t; \gamma)$ denote the distribution over states at the $n^{th}$ sequential step of a $\gamma$-model rollout beginning from state $\mathbf{s}_t$. For any desired discount $\tilde{\gamma} \in [\gamma, 1)$, we may reweight the samples from these model rollouts according to the weights*

$$\alpha_n = \frac{(1 - \tilde{\gamma})(\tilde{\gamma} - \gamma)^{n-1}}{(1 - \gamma)^n}$$

*to obtain the state distribution drawn from $\mu_1(\mathbf{s}_e \mid \mathbf{s}_t; \tilde{\gamma}) = \mu(\mathbf{s}_e \mid \mathbf{s}_t; \tilde{\gamma})$. That is, we may reweight the steps of a $\gamma$-model rollout so as to match the distribution of a $\tilde{\gamma}$-model with larger discount:*

$$\mu(\mathbf{s}_e \mid \mathbf{s}_t; \tilde{\gamma}) = \sum_{n=1}^{\infty} \alpha_n \mu_n(\mathbf{s}_e \mid \mathbf{s}_t; \gamma).$$

*Proof.* Each step of the $\gamma$-model samples a time according to $\Delta t \sim \text{Geom}(1 - \gamma)$, so the time after $n$ $\gamma$-model steps is distributed according to the sum of $n$ independent geometric random variables with identical parameters. This sum corresponds to a negative binomial random variable, $\text{NB}(n, 1 - \gamma)$, with the following pmf:

$$p_n(t) = \binom{t - 1}{t - n} \gamma^{(t-n)} (1 - \gamma)^n \tag{7}$$

Equation 7 is mildly different from the textbook pmf because we want a distribution over the total number of trials (in our case, cumulative timesteps $t$) instead of the number of successes before the $n^{\text{th}}$ failure. The latter is more commonly used because it gives the random variable the same support, $t \geqslant 0$, for all $n$. The form in Equation 7 only has support for $t \geqslant n$, which substantially simplifies the following analysis.

The distributions $q(t)$ expressible as a mixture over the per-timestep negative binomial distributions $p_n$ are given by:

$$q(t) = \sum_{n=1}^{t} \alpha_n p_n(t),$$

in which $\alpha_n$ are the mixture weights. Because $p_n$ only has support for $t \geqslant n$, it suffices to only consider the first $t$ $\gamma$-model steps when solving for $q(t)$.

We are interested in the scenario in which $q(t)$ is also a geometric random variable with smaller parameter, corresponding to a larger discount $\tilde{\gamma}$. We proceed by setting $q(t) = \text{Geom}(1 - \tilde{\gamma})$ and solving for the mixture weights $\alpha_n$ by induction.

**Base case.**  Let $n = 1$. Because $p_1$ is the only mixture component with support at $t = 1$, $\alpha_1$ is determined by $q(1)$:

$$1 - \tilde{\gamma} = \alpha_1 \binom{t - 1}{t - 1} \gamma^{t-1} (1 - \gamma)^t$$
$$= \alpha_1 (1 - \gamma).$$

Solving for $\alpha_1$ gives:

$$\alpha_1 = \frac{1 - \tilde{\gamma}}{1 - \gamma}.$$

**Induction step.** We now assume the form of $\alpha_k$ for $k = 1, \dots, n-1$ and solve for $\alpha_n$ using $q(n)$.

$$
\begin{aligned}
(1 - \tilde{\gamma})\tilde{\gamma}^{n-1} &= \sum_{k=1}^{n} \alpha_k \binom{n-1}{n-k} \gamma^{n-k}(1-\gamma)^k \\
&= \left\{ \sum_{k=1}^{n-1} \frac{(1-\tilde{\gamma})(\tilde{\gamma}-\gamma)^{k-1}}{(1-\gamma)^k} \binom{n-1}{n-k} \gamma^{n-k}(1-\gamma)^k \right\} + \alpha_n(1-\gamma)^n \\
&= (1-\tilde{\gamma}) \left\{ \sum_{k=1}^{n-1} \binom{n-1}{n-k} (\tilde{\gamma}-\gamma)^{k-1}\gamma^{n-k} \right\} + \alpha_n(1-\gamma)^n \\
&= (1-\tilde{\gamma}) \left\{ \sum_{k=1}^{n} \binom{n-1}{n-k} (\tilde{\gamma}-\gamma)^{k-1}\gamma^{n-k} \right\} - (1-\tilde{\gamma})(\tilde{\gamma}-\gamma)^{n-1} + \alpha_n(1-\gamma)^n \\
&= (1-\tilde{\gamma})\tilde{\gamma}^{n-1} - (1-\tilde{\gamma})(\tilde{\gamma}-\gamma)^{n-1} + \alpha_n(1-\gamma)^n
\end{aligned}
$$

Solving for $\alpha_n$ gives

$$
\alpha_n = \frac{(1-\tilde{\gamma})(\tilde{\gamma}-\gamma)^{n-1}}{(1-\gamma)^n}
$$

as desired. $\qquad\square$

## Appendix B  Derivation of $\gamma$-Model-Based Value Expansion

In this section, we derive the $\gamma$-MVE estimator and provide pseudo-code showing how it may be used as a drop-in replacement for value estimation in an actor-critic algorithm. Before we begin, we prove a lemma which will become useful in interpreting value functions as weighted averages.

**Lemma 1.**

$$
1 - \sum_{n=1}^{H} \alpha_n = \left( \frac{\tilde{\gamma}-\gamma}{1-\gamma} \right)^H
$$

*Proof.*

$$
\begin{aligned}
1 - \sum_{n=1}^{H} \alpha_n &= 1 - \left( \frac{1-\tilde{\gamma}}{\tilde{\gamma}-\gamma} \right) \sum_{n=1}^{H} \left( \frac{\tilde{\gamma}-\gamma}{1-\gamma} \right)^n \\
&= 1 - \left( \frac{1-\tilde{\gamma}}{\tilde{\gamma}-\gamma} \right) \frac{\left( \frac{\tilde{\gamma}-\gamma}{1-\gamma} \right) - \left( \frac{\tilde{\gamma}-\gamma}{1-\gamma} \right)^{H+1}}{\frac{1-\tilde{\gamma}}{1-\gamma}} \\
&= 1 - \left( \frac{1-\gamma}{\tilde{\gamma}-\gamma} \right) \left( \left( \frac{\tilde{\gamma}-\gamma}{1-\gamma} \right) - \left( \frac{\tilde{\gamma}-\gamma}{1-\gamma} \right)^{H+1} \right) \\
&= \left( \frac{\tilde{\gamma}-\gamma}{1-\gamma} \right)^H
\end{aligned}
$$

$\qquad\square$

We now proceed to the $\gamma$-MVE estimator itself.

**Theorem 2.** *For $\tilde{\gamma} > \gamma$, $V(\mathbf{s}_t; \tilde{\gamma})$ may be decomposed as a weighted average of $H$ $\gamma$-model steps and a terminal value estimation. We denote this as the $\gamma$-MVE estimator:*

$$
\hat{V}_{\gamma\text{-MVE}}(\mathbf{s}_t; \tilde{\gamma}) = \frac{1}{1-\tilde{\gamma}} \sum_{n=1}^{H} \alpha_n \mathbb{E}_{\mathbf{s}_e \sim \mu_n(\cdot|\mathbf{s}_t;\gamma)} \left[ r(\mathbf{s}_e) \right] + \left( \frac{\tilde{\gamma}-\gamma}{1-\gamma} \right)^H \mathbb{E}_{\mathbf{s}_e \sim \mu_H(\cdot|\mathbf{s}_t;\gamma)} \left[ V(\mathbf{s}_e; \tilde{\gamma}) \right].
$$

*Proof.*

$$V(\mathbf{s}_t; \tilde{\gamma}) = \frac{1}{1 - \tilde{\gamma}} \mathbb{E}_{\mathbf{s}_e \sim \mu(\cdot|\mathbf{s}_t; \tilde{\gamma})} \left[ r(\mathbf{s}_e) \right]$$

$$= \frac{1}{1 - \tilde{\gamma}} \sum_{n=1}^{\infty} \alpha_n \mathbb{E}_{\mathbf{s}_e \sim \mu_n(\cdot|\mathbf{s}_t; \gamma)} \left[ r(\mathbf{s}_e) \right]$$

$$= \underbrace{\frac{1}{1 - \tilde{\gamma}} \sum_{n=1}^{H} \alpha_n \mathbb{E}_{\mathbf{s}_e \sim \mu_n(\cdot|\mathbf{s}_t; \gamma)} \left[ r(\mathbf{s}_e) \right]}_{\textcircled{1}} + \underbrace{\frac{1}{1 - \tilde{\gamma}} \sum_{n=H+1}^{\infty} \alpha_n \mathbb{E}_{\mathbf{s}_e \sim \mu_n(\cdot|\mathbf{s}_t; \gamma)} \left[ r(\mathbf{s}_e) \right]}_{\textcircled{2}}. \quad (8)$$

The second equality rewrites an expectation over a $\tilde{\gamma}$-model as an expectation over a rollout of a $\gamma$-model using step weights $\alpha_n$ from Theorem 1. We recognize $\textcircled{1}$ as the model-based component of the value estimation in $\gamma$-MVE. All that remains is to write $\textcircled{2}$ using a terminal value function.

$$\sum_{n=H+1}^{\infty} \alpha_n \mathbb{E}_{\mathbf{s}_e \sim \mu_n(\cdot|\mathbf{s}_t; \gamma)} \left[ r(\mathbf{s}_e) \right] = \sum_{n=1}^{\infty} \alpha_{H+n} \mathbb{E}_{\mathbf{s}_e \sim \mu_{H+n}(\cdot|\mathbf{s}_t; \gamma)} \left[ r(\mathbf{s}_e) \right]$$

$$= \left( \frac{\tilde{\gamma} - \gamma}{1 - \gamma} \right)^H \mathbb{E}_{\mathbf{s}_H \sim \mu_H(\cdot|\mathbf{s}_t; \gamma)} \left[ \sum_{n=1}^{\infty} \alpha_n \mathbb{E}_{\mathbf{s}_e \sim \mu_n(\cdot|\mathbf{s}_H; \gamma)} \left[ r(\mathbf{s}_e) \right] \right]$$

$$= \left( \frac{\tilde{\gamma} - \gamma}{1 - \gamma} \right)^H \mathbb{E}_{\mathbf{s}_H \sim \mu_H(\cdot|\mathbf{s}_t; \gamma)} \left[ \mathbb{E}_{\mathbf{s}_e \sim \mu(\cdot|\mathbf{s}_H; \tilde{\gamma})} \left[ r(\mathbf{s}_e) \right] \right]$$

$$= (1 - \tilde{\gamma}) \left( \frac{\tilde{\gamma} - \gamma}{1 - \gamma} \right)^H \mathbb{E}_{\mathbf{s}_e \sim \mu_H(\cdot|\mathbf{s}_t; \gamma)} \left[ V(\mathbf{s}_e; \tilde{\gamma}) \right] \quad (9)$$

The second equality uses $\alpha_{H+n} = \left( \frac{\tilde{\gamma} - \gamma}{1 - \gamma} \right)^H \alpha_n$ and the time-invariance of $G^{(n)}$ with respect to its conditioning state. Plugging Equation 9 into Equation 8 gives:

$$V(\mathbf{s}_t; \tilde{\gamma}) = \frac{1}{1 - \tilde{\gamma}} \sum_{n=1}^{H} \alpha_n \mathbb{E}_{\mathbf{s}_e \sim \mu_n(\cdot|\mathbf{s}_t; \gamma)} \left[ r(\mathbf{s}_e) \right] + \left( \frac{\tilde{\gamma} - \gamma}{1 - \gamma} \right)^H \mathbb{E}_{\mathbf{s}_e \sim \mu_H(\cdot|\mathbf{s}_t; \gamma)} \left[ V(\mathbf{s}_e; \tilde{\gamma}) \right].$$

$\square$

**Remark 1.** Using Lemma 1 to substitute $1 - \sum_{n=1}^{H} \alpha_n$ in place of $\left( \frac{\tilde{\gamma} - \gamma}{1 - \gamma} \right)^H$ clarifies the interpretation of $V(\mathbf{s}_t; \tilde{\gamma})$ as a weighted average over $H$ $\gamma$-model steps and a terminal value function. Because the mixture weights must sum to 1, it is unsurprising that the weight on the terminal value function turned out to be $\left( \frac{\tilde{\gamma} - \gamma}{1 - \gamma} \right)^H = 1 - \sum_{n=1}^{H} \alpha_n$.

**Remark 2.** Setting $\gamma = 0$ recovers standard MVE with a single-step model, as the weights on the model steps simplify to $\alpha_n = (1 - \tilde{\gamma})(\tilde{\gamma} - \gamma)^{n-1}$ and the weight on the terminal value function simplifies to $\tilde{\gamma}^H$.

## Appendix C   Implementation Details

**$\gamma$-MVE algorithmic description.**   The $\gamma$-MVE estimator may be used for value estimation in any actor-critic algorithm. We describe the variant used in our control experiments, in which it is used in the soft actor critic algorithm (SAC; Haarnoja et al. 2018), in Algorithm 3. The $\gamma$-model update is unique to $\gamma$-MVE; the objectives for the value function and policy are identical to those in SAC. The objective for the $Q$-function differs only by replacing $V(\mathbf{s}_{t+1})$ with $V_{\gamma-\text{MVE}}(\mathbf{s}_{t+1})$. For a detailed description of how the gradients of these objectives may be estimated, and for hyperparameters related to the training of the $Q$-function, value function, and policy, we refer to Haarnoja et al. (2018).

---

**Algorithm 3** $\gamma$-model based value expansion

---
1: **Input** $\gamma$: model discount, $\tilde{\gamma}$: value discount, $\lambda$ : step size
2: **Initialize** $\mu_\theta$ : $\gamma$-model generator
3: **Initialize** $Q_\omega$ : $Q$-function, $V_\xi$ : value function, $\pi_\psi$ : policy, $\mathcal{D}$ : replay buffer
4: **for** each iteration **do**
5:     **for** each environment step **do**
6:         $\mathbf{a}_t \sim \pi_\psi(\cdot \mid \mathbf{s}_t)$
7:         $\mathbf{s}_{t+1} \sim p(\cdot \mid \mathbf{s}_t, \mathbf{a}_t)$
8:         $\mathbf{r}_t = r(\mathbf{s}_t, \mathbf{a}_t)$
9:         $\mathcal{D} \leftarrow \mathcal{D} \cup \{\mathbf{s}_t, \mathbf{a}_t, \mathbf{r}_t, \mathbf{s}_{t+1}\}$
10:     **end for**
11:     **for** each gradient step **do**
12:         Sample transitions $(\mathbf{s}_t, \mathbf{a}_t, \mathbf{r}_t, \mathbf{s}_{t+1})$ from $\mathcal{D}$
13:         Update $\mu_\theta$ to Algorithm 1 or 2
14:         Compute $V_{\gamma-\text{MVE}}(\mathbf{s}_{t+1})$ according to Theorem 2
15:         Update $Q$-function parameters:
$$\omega \leftarrow \omega - \lambda \nabla_\omega \tfrac{1}{2} \left(Q_\omega(\mathbf{s}_t, \mathbf{a}_t) - (\mathbf{r}_t + \tilde{\gamma} V_{\gamma-\text{MVE}}(\mathbf{s}_{t+1}))\right)^2$$
16:         Update value function parameters:
$$\xi \leftarrow \xi - \lambda \nabla_\xi \tfrac{1}{2} \left(V_\xi(\mathbf{s}_t) - \mathbb{E}_{\mathbf{a} \sim \pi_\psi(\cdot|\mathbf{s}_t)}\left[Q_\omega(\mathbf{s}_t, \mathbf{a}) - \log \pi_\psi(\mathbf{a} \mid \mathbf{s}_t)\right]\right)^2$$
17:         Update policy parameters:
$$\psi \leftarrow \psi - \lambda \nabla_\psi \mathbb{E}_{\mathbf{a} \sim \pi_\psi(\cdot|\mathbf{s}_t)}\left[\log \pi_\psi(\mathbf{a} \mid \mathbf{s}_t) - Q_\omega(\mathbf{s}_t, \mathbf{a})\right]$$
18:     **end for**
19: **end for**

---

Table 1: GAN $\gamma$-model hyperparameters (Algorithm 1).

| Parameter | Value |
|---|---|
| Batch size | 128 |
| Number of $\mathbf{s}_e$ samples per $(\mathbf{s}_t, \mathbf{a}_t)$ pair | 512 |
| Delay parameter $\tau$ | $5 \cdot 10^{-3}$ |
| Step size $\lambda$ | $1 \cdot 10^{-4}$ |
| Replay buffer size (off-policy prediction experiments) | $2 \cdot 10^5$ |

**Network architectures.** For all GAN experiments, the $\gamma$-model generator $\mu_\theta$ and discriminator $D_\phi$ are instantiated as two-layer MLPs with hidden dimensions of 256 and leaky ReLU activations. For all normalizing flow experiments, we use a six-layer neural spline flow (Durkan et al., 2019) with 16 knots defined in the interval $[-10, 10]$. The rational-quadratic coupling transform uses a three-layer MLP with hidden dimensions of 256.

**Hyperparameter settings.** We include the hyperparameters used for training the GAN $\gamma$-model in Table 1 and the flow $\gamma$-model in Table 2.

We found the original GAN (Goodfellow et al., 2014) and the least-squares GAN (Mao et al., 2016) formulation to be equally effective for training $\gamma$-models as GANs.

## Appendix D    Environment Details

**Acrobot-v1** is a two-link system (Sutton, 1996). The goal is to swing the lower link above a threshold height. The eight-dimensional observation is given by $[\cos\theta_0, \sin\theta_0, \cos\theta_1, \sin\theta_1, \frac{\mathrm{d}}{\mathrm{d}t}\theta_0, \frac{\mathrm{d}}{\mathrm{d}t}\theta_1]$. We modify it to have a one-dimensional continuous action space instead of the standard three-dimensional discrete action space. We provide reward shaping in the form of $r_{\text{shaped}} = -\cos\theta_0 - \cos(\theta_0 + \theta_1)$.

**MountainCarContinuous-v0** is a car on a track (Moore, 1990). The goal is to drive the car up a high too high to summit without built-up momentum. The two-dimmensional observation space is $[x, \frac{\mathrm{d}}{\mathrm{d}t}x]$. We provide reward shaping in the form of $r_{\text{shaped}} = x$.

Table 2: Flow $\gamma$-model hyperparameters (Algorithm 2)

| Parameter | Value |
|---|---|
| Batch size | 1024 |
| Number of $\mathbf{s}_e$ samples per $(\mathbf{s}_t, \mathbf{a}_t)$ pair | 1 |
| Delay parameter $\tau$ | $5 \cdot 10^{-3}$ |
| Step size $\lambda$ | $1 \cdot 10^{-4}$ |
| Replay buffer size (off-policy prediction experiments) | $2 \cdot 10^5$ |
| Single-step Gaussian variance $\sigma^2$ | $1 \cdot 10^{-2}$ |

**Pendulum-v0** is a single-link system. The link starts in a random position and the goal is to swing it upright. The three-dimensional observation space is given by $[\cos\theta, \sin\theta, \frac{\mathrm{d}}{\mathrm{d}t}\theta]$.

**Reacher-v2** is a two-link arm. The objective is to move the end effector $\mathbf{e}$ of the arm to a randomly sampled goal position $\mathbf{g}$. The 11-dimensional observation space is given by $[\cos\theta_0, \cos\theta_1, \sin\theta_0, \sin\theta_1, \mathbf{g}_x, \mathbf{g}_y, \frac{\mathrm{d}}{\mathrm{d}t}\theta_0, \frac{\mathrm{d}}{\mathrm{d}t}\theta_1, \mathbf{e}_x - \mathbf{g}_x, \mathbf{e}_y - \mathbf{g}_y, \mathbf{e}_z - \mathbf{g}_z]$.

Model-based methods often make use of shaped reward functions during model-based rollouts (Chua et al., 2018). For fair comparison, when using shaped rewards we also make the same shaping available to model-free methods.

## Appendix E  Adversarial $\gamma$-Model Predictions

Figure 6: Visualization of the distribution from a single feedforward pass of $\gamma$-models trained as GANs according to Algorithm 1. GAN-based $\gamma$-models tend to be more unstable than normalizing flow $\gamma$-models, especially at higher discounts.