[Reviews · NeurIPS 2020]

Review 1

Summary and Contributions: This paper proposes a new class of predictive dynamics model, called \gamma-models, which is a hybrid between model-free and model-based methods, and is closely related to successor representations. The authors present two ways to instantiate the \gamma-model: as an energy-based model and a generative adversarial network. Preliminary results show that the \gamma-model can learn long-horizon distributions accurately on simple domains, and can be incorporated into standard model-based algorithms and yields competitive results with state-of-the-art algorithms. Overall I like the idea of reinterpreting temporal difference learning as a method for training probabilistic dynamics models, especially given its ties with successor representations. The proposed approach is novel to my knowledge, and this hybrid form between model-based and model-free mechanisms is clearly relevant and of interest to the RL community.

Strengths: - A novel approach to learning predictive dynamics model. Unlike usual predictive dynamics models that models with a deterministic horizon, \gamma-model predict with a probabilistic horizon determined by \gamma. - A hybrid between model-based and model-free mechanisms, bringing together some of the favourable qualities from both worlds. - Empirical shows its potential for prediction and control.

Weaknesses: - Hard to scale up to more complex tasks, such as tasks with high-dimensional image observations. - There is always a tradeoff regarding the choice of \gamma. - Experiments are somewhat limited

Correctness: Everything looks fine.

Clarity: The paper is generally well-written and easy to follow.

Relation to Prior Work: The related work section is well-written, clearly stating the distinctions between the proposed model and prior works.

Reproducibility: Yes

Additional Feedback: - Theoretically speaking, \gamma-model should be particularly useful in a transfer learning setup. I'm wondering if you've experimented with this setup? Minor comments: Line 28: missing period after "without sequential rollouts".


Review 2

Summary and Contributions: Update: The rebuttal addresses my questions about whether it is possible to learn the model online and whether the model parameter actually controls a tradeoff. I raised the score. There is no significant improvement in control experiments but, overall, I think a generative version of successor representation is a new and important idea and the experiments in this paper just show two of the various possible applications of gamma-models. ---------------------------------------------- This paper introduces gamma-models, a generative counterpart to successor representation in reinforcement learning. An energy-based model or a generative adversarial network can be trained with a TD-style update rule to allow sampling states from a distribution which is the normalized sum of discounted probabilities of visiting a state in the future. The application of this model in planning with rollouts and control with model-based value expansion are studied.

Strengths: Successor representation is an important concept in reinforcement learning and has applications beyond the ones discussed in the paper. I believe a generative successor representation model can be of interest in other areas of reinforcement learning like exploration and temporal abstraction. The results in the submitted paper serve as a proof of concept.

Weaknesses: The experiments are limited and it is not clear to what extent gamma-models are helpful in the two applications discussed in the paper. I will elaborate on this point in the next section. Training a GAN is often hard even in supervised learning. The experiments in the paper use a pre-trained model. Although this choice is appropriately made to study the efficacy of planning with a gamma-model separately from learning the model, it is now unclear if this model can be learned online in a reinforcement learning task.

Correctness: The motivation for rollouts with gamma-models is that the model's parameter gamma (which can be separate from the environment's discount factor) controls a trade-off. Low values of gamma will require multi-step rollouts that will result in compounding errors. High values of gamma will result in bootstrap error accumulation. I did not quite understand what the latter means, and how the results in Figure 3 and Figure 4 show this trade-off. While lower gammas do require more steps for a long roll-out, the scale of compounding errors is not clear. Also, I would expect this trade-off to result in U-shaped curve in performance that is controlled by gamma and peaks at a sweet-spot. In the experiments on control, the difference between gamma-model value expansion and regular model value expansion is not significant. It seems like gamma-models do not offer any benefit in these experiments, and differences in learning curves can be explained by noise and the fact that gamma-models have one extra parameter that can be tuned. This noise can be especially high because the experiments are on neural networks and continuous action environments. Comparing gamma-models and some baselines on linear functions with tile-coded features and discrete environments can bring up the patterns and allow a more carefully controlled analysis.

Clarity: The presentation is clear to me. My only comment is that, near Eq(3), it is good to clarify if the gradient computation goes through the bootstrapped target. This does not happen in TD-style (semi-gradient) methods and I suppose this is the case in gamma-models as well.

Relation to Prior Work: The relation to prior work is made clear in the paper. The closest previous work is successor representation (SR) in tabular domains, and it is shown that gamma-models and SR are equivalent in these domains.

Reproducibility: Yes

Additional Feedback:


Review 3

Summary and Contributions: The paper proposes a novel algorithm for training generative models of the environment---\gamma-models. These models can generate future states without having to unroll rollouts iteratively for many steps. This approach draws inspiration from the successor representation as it aims to sample future states from the discounted state-vistation distribution for the current policy. Theoretical connections are made between the successor representation and \gamma-models, leading to a TD-like training procedure. Empirical experiments examine the utility of this approach compared to one-step models and find that the proposed model can improve performance on standard control benchmark tasks.

Strengths: - The paper tackles a fundamental problem for model-based RL: How to generate predictions of future states without unrolling a model with compouding errors? The paper proposes a novel method for tackling this which could be of interest to the community. - The usage of the Bellman equation over state-visitation distributions to learn a generative model is a nice technique which I have not seen previosuly. The mechanism for using rollouts generated from one \gamma-model to simulate samples from another model with different \gamma also seemed interesting. - The control experiments show the benefits of using these models as opposed to fixed-horizon models or model-free methods.

Weaknesses: - One weakness of the successor representation is that it is policy-dependent. So, in the control setting, it would need to be relearned whenever the policy is modified. \gamma-models would also seem to suffer from the same issue. On the other hand, perhaps one-step models would not suffer from this problem (since they are conditioned on actions too). Could you comment on this issue? - If I understand correctly, \gamma-models generate states from the discounted state-visitation distribution. So, it would seem like, when the model outputs a prediction, the agent would not know how far into the future this state is---it could be the very next state or far into the future. Intuitively, this seems problematic because the time horizon of a prediction should be important in assessing its usefulness to the agent. For example, knowing that you will crash a car in 10 seconds if you continue straight could be important, but the model might generate a prediction that is only 1 second in the future. In other words, while the model may be able to produce "jumpy" predictions, it does not seem to choose the "jumps" in a particularly useful way, which sheds some doubt on the utility of this approach. - In Fig. 4 b), we see that the errors for the \gamma-model and an iterated 1-step model are very similar, yet, in the control experiments \gamma-MVE outperforms MVE. Why do you think that is the case? The previous experiment suggests that errors are similar for both. - For the control experiments, it would be nice if more seeds were used, particularly for the simpler environments like Mountain Car. Currently, the standard error bars are relatively large and only 5 runs are done, making it difficult to assess the reliability of the conclusions. - Why are energy-based models introduced? In the end, it seems like a GAN-like training procedure is used and I am not sure if mentionning energy-based models adds to the discussion. It would seem simpler to talk about a generic parameterized distribution at the beginning of section 4 instead.

Correctness: Generally, the methodology seems sound. See "Weaknesses" for other suggestions.

Clarity: Overall, the paper is well-written. The clarity of certain parts could be improved. Some points to consider: - I found Figure 2 difficult to understand. Could you clarify these? In particular, in the last sentence of the caption, I am not sure why a certain number of steps is required for reweighting to be possible. - What is Figure 3 trying to show? I am confused whether the plots are the outputs of a learned model or Monte Carlo estimates as indicated by the caption. - Equation 2 was unclear to me. Should the quantity on the left-hand side be G(s_e|s_t, a_t)? Currently, it does not seem like a valid Bellman equation because there is no recursive quantity.

Relation to Prior Work: While a lot of the prior work is mentioned in the introduction, I think certain parts could be elaborated on: - Are there any works that attempt to sample from the state-visitation distribution? I was not exactly sure if this was part of the contributions of this paper. - There are other works that directly learn multi-step models with fixed horizons i.e. n-step models. How do different fixed horizons compare to using different value of \gamma for prediction and control? I think it would be nice to discuss the advantages of \gamma-models over fixed-horizon models a little more.

Reproducibility: Yes

Additional Feedback: I would be willing to revise my score based if my concerns are addressed.


Review 4

Summary and Contributions: This paper reinterprets temporal difference learning as a method for training generative models for environment dynamics and develop a gamma-model for prediction with probabilistic horizon. It addresses the compounding error problem in MBRL (caused by long-horizon rollouts) by dealing it during the training stage (error caused by bootstrapping). The paper develops a practical GAN version of the gamma-model, which learns to directly generate samples of states. The method could be integrated with existing model-free algorithms by replacing the value functions therein by the ones estimated by the gamma-model. Experiments shows that the approach could improve over existing method. It also extensively examines error compounding problem in experiments and compares to standard single-step model.

Strengths: The paper provides a new interpretation of model-learning in MBRL as a problem of learning generative model, which could leverage a rich set of recent works (e.g., GAN). It provides an alternative approach to dealing with the compounding error problem. The proposed method is principled and elegant.

Weaknesses: All the experiments are carried out on toy benchmarks. Applying it to more complex domains such as Atari games remains challenging due to the intrinsic difficulty of learning high-dimensional generative models. Therefore, it remains unclear whether it is still competitive compared to other model-free and model-based methods on these more challenging tasks.

Correctness: Yes.

Clarity: Yes.

Relation to Prior Work: Yes.

Reproducibility: Yes

Additional Feedback: 1. It would be more significant if the paper could show at least some evidence that the proposed method is also promising on high-dimensional domains (such as Atari). 2. Currently in experiments, the proposed gamma-model is used to estimate the value function and plugged into soft actor-critic. It would also be more compelling if the gamma-model be used for planning actions. (For example, could the learned gamma-model be combined with MCTS for action planning? And how?)

[Author Response · NeurIPS 2020]

We thank the reviewers for their insights and suggestions. Questions primarily concerned the utility of $\gamma$-model
predictions and associated tradeoffs, policy-dependent models, and scalability. We clarify how $\gamma$-model predictions
are used for decision-making (with further elaboration on value estimation), discuss how the tradeoff induced by the
choice of $\gamma$ is one that must be made in any RL algorithm, and comment on scaling to higher-dimensional environments.
Answers below will be included in expanded discussions in future versions of the paper.

**(R3) Utility of "jumpy" predictions.** One of the insights of this paper is that model-based RL algorithms do not
need to know the time at which a state will be encountered as long as states are sampled according to the discounted
occupancy. This does mean that decisions should be based on expectations over many samples (Figure R1b) and not
based on a single sample. In the case of R3's car example, as long as states from 10 steps into the future are sampled
according to a probability of $p(t = 10) = (1 - \gamma)\gamma^9$, crashing at $t = 10$ will be reflected in such an expectation.

**(R2, R3) Clarifications on RL experiments.** (i) In RL experiments, $\gamma$-models were **not** pre-trained; they were trained
online. (ii) Though bounds exist, it is difficult to relate control performance to state prediction accuracy in practice
[Lambert et al., 2020]. The intermediate quantity of value prediction error (Figure R1a) is more informative in this
respect, and shows a small accuracy improvement for $\gamma > 0.95$, consistent with the performance of $\gamma$-MVE vs MVE.

**Figure R1:** **(a)** Value prediction error for an occupancy with target discount $\widetilde{\gamma} = 0.99$ as a function of model discount $\gamma$. Higher $\gamma$'s perform slightly better. **(b)** Visual depiction of value estimation (in Pendulum) as an expectation of reward over $\gamma$-model predictions.

**(R1, R2) Tradeoff with $\gamma$.** The $\gamma$-model bridges the gap between model-based and model-free learning, and as such
allows one to trade off between training-time model-free errors (referred to as bootstrap error accumulation; Kumar et al.
2019) and testing-time model-based rollout errors. The tradeoff between these two is not a weakness of the $\gamma$-model,
nor is it even unique to the $\gamma$-model; such compounding errors are unavoidable in sequential decision-making. The
advantage of the $\gamma$-model is that it allows for more careful interpolation between these extremes.

**(R2, R3) Policy-dependence.** The $\gamma$-model is policy-dependent in the same way that a Q-function is. While the single-
step transition distribution is certainly independent of a policy, parametric single-step models are still policy-dependent
because the policy determines the training data distribution for statistical learning.

**(R1, R4) Scaling to image-based environments.** We agree that experiments in more complex domains would improve
the submission. Unfortunately, model-based experiments in image domains like Atari games are quite complex to set
up, generally requiring specialized image prediction models that can take weeks to train [Kaiser et al., 2019]. We have
found that $\gamma$-model training scales to state spaces of the MuJoCo gym benchmark suite for discounts up to $\gamma = 0.9$, but
leave extensions to image-based environments for future work.

**Other questions from Reviewer 2**   **❶** *Gradients:* The gradient computation indeed does not pass through the
bootstrapped target. This is discussed in L211-L215 in Section 6 "Practical Training of $\gamma$-Models".

**Other questions from Reviewer 3**   **❶** *Prediction visualization:* The first five columns of Figure 3 depict learned
$\gamma$-model predictions. The only Monte Carlo trajectory estimates are in the final column for comparison.   **❷** *Relation*
*to $n$-step models:* Training an $n$-step model for $n$ larger than 10 or 20 quickly becomes impractical, and is impossible
without on-policy samples. By directly learning the discounted occupancy, we are able to train a model with much
larger effective (probabilistic) horizon while only using off-policy single-step transitions.   **❸** *Bellman equation:* There
is no recursion in Equation 2 because it describes a bootstrapped target distribution (akin to a bootstrapped target value
$r(\mathbf{s}_t) + \gamma V(\mathbf{s}_{t+1})$) and is not a Bellman equation by itself.   **❹** *Required rollout lengths:* Figure 2b shows the length
of a $\gamma$-model-based rollout required to recover $95\%$ of the probability mass of an occupancy with discount $\widetilde{\gamma}$. This is
more precise than an effective horizon. For example, the effective horizon for $\widetilde{\gamma} = 0.99$ with a single-step model is 100,
which recovers only $1 - .99^{101} \approx 64\%$ of the full infinite-horizon occupancy.   **❺** *Energy-based model:* We introduce
the EBM formulation because it provides the simplest way to explain how temporal difference updates may be used to
train generative models. A neural generator reduces training times, but this is more an implementation choice than a
fundamental component of infinite-horizon prediction, so we opt to first explain the framework with as few moving
pieces as possible. Similarly, we have found that the $\gamma$-model may also be instantiated as a normalizing flow, and will
add experiments comparing the flow and GAN instantiations of $\gamma$-models.

**References**   N Lambert et al, *arXiv:2002.04523*. A Kumar et al, *arxiv:1906.00949*. L Kaiser et al, *arXiv:1903.00374*.

[Meta-Review · NeurIPS 2020]

Summary: this paper proposes a new model-based RL algorithm, where instead of learning state transition probabilities, the occupancy distribution for an infinite horizon is learned. This method can be seen as an extension of the method known as the successor representation to continuous state-action spaces and to infinite horizons. The occupancy distribution is modeled as an energy function, and learned with temporal differences (TD), using a GAN. The experiments on a few MuJuCo problems clearly show the advantages of the proposed approach compared to RL algorithms such as PPO and SAC. The reviewers agree that the proposed method is new, interesting, and validated by the simulation experiments. There are some concerns about the limitation of the experiments and scalability to high dimensional observations such as images where learning occupancy distributions is clearly challenging. The fact that the occupancy is policy-dependent is also a major limitation.